# Gender, skin color, and household composition explain inequities in household food insecurity in Brazil

**Lissandra Amorim Santos**[1], **Rafael Pérez-Escamilla**[2]⊙*, **Camilla Christine de Souza Cherol**[1‡], **Aline Alves Ferreira**[1‡], **Rosana Salles-Costa**[1]⊙

**1** Department of Social and Applied Nutrition, Institute of Nutrition Josué de Castro, Federal University of Rio de Janeiro, Rio de Janeiro, Rio de Janeiro, Brazil, **2** Department of Social and Behavioral Sciences, Yale School of Public Health, New Haven, Connecticut, United States of America

⊙ These authors contributed equally to this work.
‡ These authors also contributed equally to this work
* rafael.perez-escamilla@yale.edu

**Data Availability Statement:** All the databases used are secondary data of unidentified public domain data collected through national household sample surveys. The POF 2017-8 dataset is available in the URL: https://www.ibge.gov.br/

## Abstract

It is well known that female-headed households (FHHs) are more likely to experience food insecurity (FI) than male-headed households (MHHs), however there is a dearth of evidence on how gender intersects with other social determinants of FI. Thus, this paper investigated changes in the prevalence of household FI in Brazil from 2004 to 2018 by the intersection of gender, race/skin color and marital status of the household reference person. Data from three cross-sectional nationally representative surveys that assessed the status of FI using the Brazilian Household Food Insecurity Measurement Scale were analyzed ($N_{2004}$ = 107,731; $N_{2013}$ = 115,108, $N_{2018}$ = 57,204). Multinomial logistic regression models were used to examine the relationship between profiles of gender, race/skin color, marital status of the head of the household with household FI stratified by the presence of children <5 years of age. Over time, FHHs had a higher prevalence of mild and moderate/severe FI than did households headed by men. Food security prevalence increased from 2004 to 2013 and decreased between 2013 and 2018 for households headed by men and women. In 2018, households headed by black/brown single mothers with children < 5 years of age were at the highest FI risk. The probability of reporting moderate/severe FI in these households were 4.17 times higher (95% CI [2.96–5.90]) than for households headed by married white men. The presence of children in the household was associated with a higher probability of moderate/severe FI, especially for households headed by black/brown individuals regardless of the reference person's gender. The results suggest that gender inequities combined with darker skin color and the presence of children at home potentiate the risk of moderate/severe FI. Policy makers need to consider the principles of intersectionality when investing in codesigning, implementing, evaluating, and scaling up evidence-based programs to reduce FI.

estatisticas/sociais/saude/24786-pesquisa-de-orcamentosfamiliares-2.html?=&t=microdados. Data from the PNAD database for the years 2004 and 2013 are available in the URL: https://www.ibge.gov.br/estatisticas/sociais/educacao/9127-pesquisa-nacional-poramostra-de-domicilios.html?=&t=microdados.

**Funding:** This study was financed in part by the Coordenação de Aperfeiçoamento de Pessoal de Nível Superior – Brasil (CAPES) – Finance Code 001. LAS received financial support by Coordination for the Improvement of Higher Education Personnel (Coordenação de Aperfeiçoamento de Pessoal de Nível Superior - CAPES) (Grant: 88887.511179/2020-00). The research group (GISAN - UFRJ), coordinated by RS-C, is supported by The Brazilian National Council for Scientific and Technological Development (Conselho Nacional de Desenvolvimento Científico e Tecnológico - CNPq) (Edital Universal, Grant 423174/2018-5) and Foundation Carlos Chagas Filho Research Support of the State of Rio de Janeiro (Fundação de Amparo à Pesquisa do Estado do Rio de Janeiro - FAPERJ) (Edital APQ1 2019, Grant E-26/10.001596/2019). CAPES, CNPq or FAPERJ had no role in the study design, analysis, decision to publish, or preparation of the manuscript.

**Competing interests:** The authors have declared that no competing interests exist.

## 1. Introduction

The State of Food and Nutrition Security in the World (FNS) reported that there were 2.37 billion people suffering from food insecurity (FI) in 2020, and this number remained mostly unchanged in 2021 [1]. The prevalence of FI was higher among women than men, and the gender gap in FI increased, especially in the previous two years, due to the economic crisis triggered by the COVID-19 pandemic. Latin America and the Caribbean are some of the regions most affected by the widening FI gender gap (9.4% in 2020 vs. 11.3% in 2020) [1].

The FAO defines FI as a condition that exists "when people lack secure access to sufficient amounts of safe and nutritious food for normal growth and development and an active and healthy life" [2]. The definition of FI in Brazil also considers the potential psychosocial consequences of this condition, including the feelings of social exclusion, low self-esteem, stress, and emotional suffering [3].

In the early 2000s, Brazilian households were experiencing major FI challenges and this situation motivated a series of social movements that culminated with the development of the Brazilian Food Insecurity Scale [*Escala Brasileira de Insegurança Alimentar* (EBIA)] in 2004 [3,4]. Between 2004 and until 2013, there were substantial reductions in FI rates in Brazil but since 2016 this trend reversed. Household FI worsened dramatically as a result of a political and economic crisis that strongly undermined the food security (FS) policies that were successfully implemented during the previous political administrations [5].

Women play a central role in household FS due to the household activities that they have historically been responsible for, including caregiving, which involves protecting the health and food security of children and other family members [6–8]. This is challenging for women as they usually have lower social capital, and experience substantial barriers related to gender norms to accessing healthy foods [9,10]. Despite this, women have lower incomes than men while having a double or even triple workload because of their additional household caregiving activities [6,11].

Women's vulnerability to FI has also been addressed in debates about poverty and gender [12–15]. Consistent evidence has shown that the poorer the household is, the more compromised its access to food or a healthy and nutritious diet becomes for all household members which shows why FI is so important for improving public health [16,17].

Furthermore, gender intersects with other social determinants of health compounding women's social vulnerabilities. For example, the intersection of female gender with black/brown race and marital status, increases substantially the risk of FI. It is well known that households headed by indigenous and black people around the world, are more likely to report FI when compared to households headed by whites or the dominant race [12,14,16,18]. It is also well known that the lifetime accumulation of social and economic disadvantage among black people is associated with a substantial increase in the risk of FI within this population [19]. Finally, non-white women rearing children or adolescents as single mothers are more likely to report FI [20] and have worse living conditions than those living in households with other family structures, regardless of gender [21]. When there are children living under FI conditions, women's situation can be even worse because mothers are usually more engaged in attempting to prevent hunger among their children, including forgoing their own FS to the benefit of their offspring [7,22].

Although it is well known that households headed by women are more likely to experience FI, there is scarce evidence on how gender intersects with other factors, such as race or ethnicity, marital status, and the presence of young children at home, to determine FI risk [14,18,23,24]. The intersectional approach was coined by black feminists and emphasizes that social structural systems of racism, patriarchy, classism, and others are not experienced

independently, but rather, they interact with each other and mutually reinforce one another [25–27].

Based on the concept of intersectionality, we hypothesize that the risk of FI will be stronger when both female gender and dark skin color are presented simultaneously than when either is presented alone [26–28]. We also anticipate that the FI risk will be compounded if black women are also household heads and when there are children living in the household.

Thus, the aim of this study is to investigate changes between 2004 and 2018 in the proportion of Brazilian households experiencing FI by the gender of the reference person and its intersection with her race/skin color, marital status and household composition. The 2004–2018 period was chosen because of the availability of national representative surveys that assessed FI using EBIA. Since the COVID-19 pandemic exploded in early 2020, Brazil experienced an additional worsening of FI leading to a sharp increase in the number of hungry people. Hence, the analytical methodology developed for this study to understand FI intersectionality during the pre-pandemic scenario can then be replicated in the context of the pandemic once nationally representative data becomes available from Brazil and other countries.

## 2. Data

These repeated cross-sectional analysis studies used data from three nationally representative surveys assessing FI in Brazil: two National Household Sample Surveys (*Pesquisa Nacional por Amostra de Domicílios–PNAD* [2004 and 2013]) and the Family Budget Survey (*Pesquisa de Orçamentos Familiares–POF* [2018]). All three national surveys included population health information and were carried out by the Brazilian Institute of Geography and Statistics (*Instituto Brasileiro de Geografia e Estatística–IBGE)*. Additionally, all three surveys collected face-to-face data by well-trained and supervised staff, including robust data quality control measures [29].

### 2.1 Samples

All household surveys conducted by IBGE, including PNADs and POF, use the same sampling design based on IBGE's master sample known as the "Integrated System of Household Surveys" (*Sistema Integrado de Pesquisas Domiciliares—SIPD*), obtained from 2010 Demographic Census data. This sampling system is based on a set of primary sampling units (PSUs), in which the survey collects data on common variables using well-standardized procedures for collecting and listing census sectors [29]. PSUs were selected based on a probability proportional to the size of the cluster according to the number of private households per census tract.

The PNADs were carried out using a probabilistic sample of households obtained in three stages of selection, selecting municipalities in the first stage, census tract in the second and households in the third stage. On the other hand, the POF survey adopted a stratified, two-stage probabilistic cluster sampling design with the selection of census tracts as PSUs in the first stage and households in the second stage.

PNAD surveys are applied during three months of the year, while POF is applied during the whole year, but access to food in Brazil is more influenced by income, rather than seasonal variations in food intake [30,31], hence this is unlikely to have affected the comparability across surveys. Even though the two surveys used slightly different approaches to household selection, the common sampling framework and just focusing on permanent private household ensured that comparability between both surveys, which are representative at the national and macroregional levels as well as urban and rural areas.

The IBGE surveys classify the area of residence as urban or rural the same way as the Brazilian demographic census does. Urban areas were defined as areas corresponding to cities

(municipal seat), towns (district seat) or isolated urban areas with more than 50,000 inhabitants in a densely populated area. By contrast, rural areas were those towns with no more than 3,000 inhabitants, or those outside the urban areas [32].

The PNAD surveys are applied to different types of households, including permanent private households and collective living households, such as hotels, pensions, orphanages, or asylums. By contrast, the POF survey only included permanent private households. To ensure the comparability between POF and PNAD surveys, in the case of PNAD we selected only the permanent private households for the analyses. Thus, in our analyses both surveys are comparable in terms of sampling design and methodology to estimate food security and FI in permanent private households. More specific details on the surveys' methods and procedures can be found elsewhere [29,33,34].

All survey participants who responded to EBIA on behalf of the household (usually the heads of household) were included in our analyses. However, it is important to note that, for both surveys, the interviewers were instructed not to accept a person under 14 years of age as an informant. Furthermore, households headed by individuals who self-identified as being "yellow" and indigenous were excluded because of their lack of representativeness in the survey (0.6% in 2004, 0.9% in 2013, and 1.2% in 2018). Among those responding to EBIA a very small percent had missing information in any of the variables of interest (1.4% in 2004;3.7% in 2013; no missing data in 2018), hence missing data was not an issue in our study. Thus, the final sample size in each survey analyzed was 107,731 (PNAD 2004), 115,108 (PNAD 2013), and 57,204 (POF 2018).

## 2.2 Assessment of household food insecurity

Household FI, the primary outcome in our analyses, was measured with the Brazilian Household Food Insecurity Measurement Scale (*Escala Brasileira de Insegurança Alimentar*—EBIA), which has been repeatedly validated since 2004 for its construct, face, internal consistency, predictive and concurrent validity (in our study EBIA's Cronbach's alpha ranged from 0.91 to 0.94). As a result, EBIA is a well validated robust experience-based scale to assess household FI in rural and urban households in Brazil [4,35,36].

EBIA is an experienced-based scale adapted from the US Household Food Security Survey Module (HFSSM) [4,37]. The scale has been used repeatedly in national surveys in Brazil since 2004 and has been a valuable tool for informing FNS governance in the country (Pérez-Escamilla 2012) [38]. Since its introduction in the 2004 PNAD, the EBIA has undergone adaptations in the number of items used to evaluate food security/FI levels, with no loss of the comparability of estimated prevalence across surveys [39]. Therefore, findings from such surveys can help target and explain the impact of policies and programs designed to mitigate FI in Brazilian families [3,38]. EBIA is based on the experience with FI, including hunger, among household members as reported by a household member who is knowledgeable of the food situation in the family. In both surveys, POF and PNAD, the scale was answered by a person within the family responsible for purchasing and preparing meals was the preferred interviewee. The time frame used to capture the FI experience was the three months preceding the survey. The current version of the EBIA consists of 14 questions with dichotomous answer options ("yes" or "no"). Eight items apply only to adults in the households (aged 18 years or above), and 6 items apply exclusively to households with children and/or adolescents (aged 18 years or less). Each household is assigned an additive score based on the total number of EBIA items affirmed. The score is then coded into three categories based on standard cutoff points to classify each household as food secure (0 point with or without people under 18 years of age), mildly FI (only adults: 1–3 points; with children: 1–5 points), moderately FI (only adults:

4–5 points; with children: 6–9 points), or severely FI (only adults: 6–8 points; with children: 10–14 points) [39].

## 2.3 Intersectional factors

The intersectional factors evaluated included characteristics of the head of the household, such as sex, race/skin color and marital status, and the presence of children less than 5 years old in the household.

In the PNAD surveys, the head of household, or the reference person of the household, was identified by the members of the household and in the POF survey it was considered by the interviewers as the individual who contributes the most to household income. It is important to note that usually the survey respondent was the reference person of the household, but when they were absent the oldest person in the household was interviewed [29,33,34].

Sex was collected as a dichotomous variable (male or female) but since the discussion on this paper will mainly address the behavior of men and women based on the social constructs of gender, and gender relations, it will be used 'gender' to discuss the topics related to it [40]. In the PNAD, marital status was captured through a six-category variable (single, divorced, separated, widowed, married or cohabitating) and reclassified into two categories (married = people cohabiting with a partner or who were married; single = single, divorced, separated, or widowed people). In POF, marital status was determined by the response to one question that evaluated whether the reference person lived alone or with a partner.

In Brazil, racial identification is usually based on self-identification following phenotype rather than descent [41,42]. For both surveys, the self-reported variable race/skin color was collected using five options (white, black, brown, yellow, or indigenous). The proportion of people who self-identified as black (based on skin color) ranged between 7.5% and 11.4% across surveys, and the proportion of people who considered themselves to be brown (*pardos*, in Portuguese) ranged between 42.7% and 49.3%. Both racial categories were combined because Brazilian race dynamics are "ambiguous and fluid" and the boundaries between black and *pardos* are blurry as both are Afro-descendant and experience systematic racism and discrimination [41]. Furthermore, the Brazilian black movement highlights the political importance of representing the Afro descendant population as the people self-identifying as black or brown based on their skin color [41,42]. Thus, for this study, the skin color variable was further reclassified into two categories (white or black/brown).

The following persona profiles were created based on the intersection of gender with race/skin color and marital status: single or married 'white women', single or married 'black/brown women', single or married 'white men' and single or married 'black/brown men'. The presence of children under 5 years of age in the household was also evaluated ('Yes' or 'No'), calculated based on the age information of the household members for both surveys. These patterns were selected a priori based on having been consistently identified as independent sociodemographic risk factors for FI [16,20].

## 2.4 Covariates

Sociodemographic characteristics were evaluated for the head of household through similar modules in the PNAD and POF surveys. Based on the conceptual FS determinants model proposed by Kepple and Segall-Correa (2011) [3], the following household characteristics were chosen: region of residence (North, Northeast, Southeast, Midwest, South); household area (urban, rural); number of residents (1–2, 3–5 or >5); education level (illiterate, 1–7 years, 8–12 years, and >12 years); and the monthly per capita household income (ratio of the sum of all family income and the number of family members) by quintiles.

## 2.5 Statistical analyses

All survey data were analyzed taking into account the complex survey designs. The prevalence and corresponding 95% confidence intervals (95% CI) of different levels of FS and FI were estimated by gender, survey year and macro region of Brazil. In this study, the EBIA classification was modeled as a three-level variable (FS, mild FI, moderate/severe FI). Chi-square tests were used to compare the categorical variables by gender and survey year. Then, the prevalence of moderate/severe FI was estimated by intersectional factors, e.g., 'white women', 'black/brown women', 'single black/brown women' and 'single black/brown women with children'.

Multinomial logistic regressions were conducted for each of the three surveys to estimate the prevalence of mild and moderate/severe FI across the intersection of gender, race/skin color and marital status of the head of the household and presence of children less than 5 years old in the household. First, unadjusted bivariable logistic multinomial regression models were evaluated for each persona profile of the head of household and the covariates. All the covariates that yielded a p value below 0.20 were included in the adjusted models. Household characteristics such as monthly per capita household income, macro region, area, and number of residents in the household were identified as adjustment factors in the models. Effect modification by the presence of children under 5 years of age in the household was examined through the testing of statistical interactions. In the third step, all models were run by macroregion or stratified by the presence of children less than 5 years of age in the household adjusted for the adjustment variables. Married white men had the lowest prevalence of moderate/severe FI and one of the highest prevalence of FS in all survey years, this is why it is considered the reference profile. The data were expressed as relative risk ratios (RRR), computed by exponentiating the multinomial logistic regression coefficients, and the corresponding 95% confidence intervals (95% CIs).

The probabilities of each profile reporting moderate/severe FI were predicted for each year after multivariable multinomial logistic regression. The '*survey*' command of Stata version 16.1 was used to account for the complex sample design (StataCorp LLC, College Station, 2016).

## 2.6 Ethical considerations

IBGE's activities and surveys are governed by Law N˚ 5,534, of November 14, 1968, that requires the government to provide access to the public to the survey data and information. According to this Law, all individuals and legal entities are guaranteed confidentiality while at the same time requiring them to answer the surveys administrated by IBGE, with reassurances that all information provided will be used exclusively for deidentified statistical analysis purposes. Additionally, the secondary data analyses presented in this article are based on public domain information and did not require additional IRB approval according to Resolution N˚ 510 of April 07, 2016, from the National Committee of Ethics in Research (CONEP). This resolution indicates that approval by local and national Ethics Committee (CEP-CONEP System) is not needed for researchers who use secondary data available in the public domain or from data made available in the public domain by the IBGE.

## 3. Results

### 3.1 Survey sample characteristics

Food security prevalence increased between 2004 and 2013 and decreased in 2018 for households headed by both men and women. Compared to male-headed households (MHHs), FHHs had a higher prevalence of both levels of FI (mild and moderate/severe) and a lower

prevalence of FS in all surveys. However, the gender gap in 2013 was lower than that in 2004 and 2018 (Fig 1).

As expected, in Brazil, most households included in the surveys were located in urban areas. Importantly, households headed by women were more likely than households headed by men to have at least 5 people living in them (6.2% vs. 5.0%). In 2018, among the FHHs, 61.5% of women were single, in contrast to the 18.7% of MHHs in which men were not married, and 15.2% of FHHs had children less than 5 years old (vs. 17.5% for MHHs) (Table 1).

Over time, the proportion of households headed by women increased and reached 41.8% in 2018. Families with female heads of household had a higher prevalence of having a head of household with more than 12 years of formal education (21.4%) than households headed by men (18.2%) (Table 1). However, despite their higher levels of education, women who were heads of their households were still significantly less likely to have incomes falling in the two upper income quintiles (20.6% and 23.8%, respectively) compared to households headed by men (25.9% and 21.4%, respectively).

Compared to other regions, the northeast region had the highest prevalence of FHH, as well as a lower disparity in relation to the prevalence of MHH in this region. Additionally, there were regional differences in the proportions of MHHs and FHHs (S1 Table).

## 3.2 Prevalence of moderate and severe FI across time by intersectional factors

Fig 2 shows how the joint consideration of gender, race/skin color, marital status of the household head, as well as the presence of children in households, provides a more nuanced understanding of which profile is most likely to be FI (numbers are provided on S3 Table). For instance, compared to the 'white married' profile (gray bar), women had higher FI prevalence across survey years. When the heads of the household declared themselves to be black or brown (based on skin color) FI prevalence increased even more. Following a stepwise trend, FI increased after adding that the household was single headed, and then when adding the presence of children younger than 5 years old in the household (Fig 2).

Importantly, the increase in percentage points by profile differed over time, and between men and women. In 2004, 13.0% (95%CI 11.5–14.6) of the households with moderate/severe FI were headed by married white women, and 28.8% (95%CI 26.9–30.9) were headed by married black/brown women. From 2013 to 2018, there was an increase of 7.2 percentual points (pp.) in the prevalence of moderate/severe FI in black/brown married women-headed households (2013: 11.1 [95%CI 10.3–12.0]; 2018: 18.3 [17.0–19.7]). In 2018, in households headed by black/brown single-mother, the prevalence of moderate/severe FI was 19.6 pp. higher than for households headed by married white women while the increase was of 13.2 pp. for the same male profiles.

Notably, MHHs had the largest FI increases with the added factors in 2004, especially when the household was headed by single black/brown men who had children less than 5 years old but the disparities between the profiles was lower in 2018 (Fig 2). Additionally, the inequity in prevalence between the profiles decreased over time, even though the prevalence of moderate/severe FI was always higher in households headed by black/brown single men and women with children under 5 years old, compared to the rest of profiles.

## 3.3 FI by gender, race, marital status, and presence of children in the household

Over time, the highest probability for experiencing mild and moderate/severe FI was for those households headed by single black/brown women, compared with households headed by

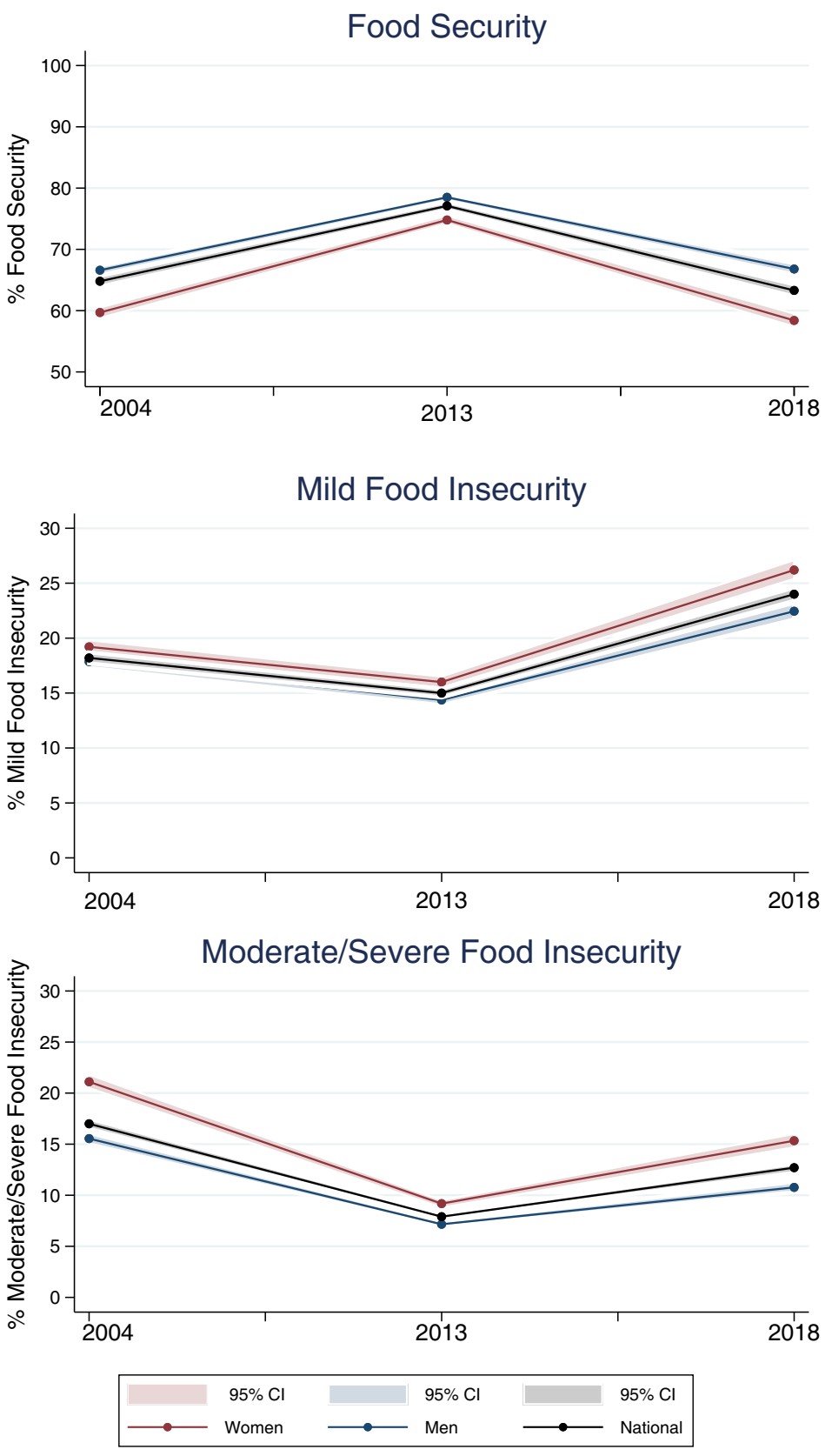

**Fig 1. Prevalence (%) of food security (FS) and levels of food insecurity (FI) in Brazilian households according to the sex of reference person and survey year.** Brazil, 2004, 2013 and 2018.

married white men. For this profile, the difference was even higher in households with children less than 5 years of age (Fig 3).

Households headed by single black/brown women were three times more likely to report moderate/severe FI in 2004 and 2013 and four times more likely to do so in 2018. The presence of children less than 5 years old in the households increased the probabilities of reporting moderate/severe FI in all surveys compared to those without children younger than 5 years old (3.34 [95%CI 2.82–3.86] vs. 3.21 [95%CI 2.92–3.50] in 2004; 3.67 [95%CI 2.93–4.41] vs. 2.91 [95%CI 2.55–3.27] in 2013; and 4.17 [95%CI 2.73–5.61] vs. 3.86 [95%CI 3.28–4.49] in 2018, respectively).

In all surveys, households headed by black/brown single men were more likely to report moderate/severe FI than households headed by white single women compared to the reference group, married white men. Compared to the reference profile, the risk ratio for moderate/severe FI were two times higher in households where black/brown single men were heads of household and when the household had at least one child under 5 years old (RRR: 2.13 [95%CI 0.51–3.75]) in 2018. Similarly, households headed by single white women were almost twice as likely to report moderate/severe FI (RRR: 1.82 [95%CI 0.95–2.69]) than those headed by married white men (Fig 3).

Single women were more vulnerable than married women to experiencing moderate/severe FI regardless of race/skin color, although these differences were larger for black/brown women. In 2018, married white women had 1.35 (95%CI [0.68–2.02]) higher probability of reporting moderate/severe FI than the reference profile in households with children less than 5 years old compared to 1.82 (95%CI [0.95–2.69]), for single white women. Meanwhile, in 2018, married black/brown women, who were heads of households with children less than 5 years old, were twice as likely to live in moderate/severe food insecure households (RRR: 2.08 [95% CI 1.38–2.78]) than the reference group, white married men, compared those with heads of household who were single and had four times the odds than the reference profile (RRR: 4.17 [95%CI 2.73–5.61]) (Fig 3).

Compared to those without children younger than 5 years old, the presence of young children in the households increased the probabilities of reporting both mild and moderate/severe FI in all surveys for most profiles. This increase was more pronounced for black/brown single men. In 2004, the FI RRR's were 2.61 (95%CI 2.30–2.92) in households without children vs. 3.34 (95%CI 1.95–4.73) in households with children. In 2013 the corresponding estimates were 2.68 (95%CI 2.31–3.05) vs. 2.92 (95%CI 2.03–3.81), respectively.

For black/brown married women, however, the probabilities of reporting moderate/severe FI were lower in households with children less than 5 years old compared to those without them. In 2004, the FI RRRs were 2.60 (95%CI 2.02–3.18) vs. 3.17 (95%CI 2.65–3.69). The corresponding estimates in 2013 were 2.05 (95%CI 1.64–2.46) vs. 2.47 (95%CI 2.12–2.82), respectively, and in 2018 they were 2.08 (95%CI 1.38–2.78) vs. 2.72 (95%CI 2.27–3.18), respectively.

There was also a strong between-region difference, with the probability of reporting moderate/severe FI being substantially higher in the southeast and southern regions than in the remaining regions (S2 Table), despite the lower prevalence of this level of FI in these regions (S1 Table).

## 4. Discussion

Our results documented how household FI vulnerability was a function of gender, race/skin color, marital status, children living in the household, and most importantly, the combination

**Table 1. Sociodemographic profile stratified by sex of the reference person according to national surveys.** Brazil, 2004, 2013 and 2018.

| | PNAD[1] 2004 | | PNAD[1] 2013 | | POF[2] 2018 | |
|---|---|---|---|---|---|---|
| | Women % (CI 95%) | Men % (CI 95%) | Women % (CI 95%) | Men % (CI 95%) | Women % (CI 95%) | Men % (CI 95%) |
| **Reference Person** | | | | | | |
| **Sex** | 26.0 (25.6–26.3) | 74.0 (73.7–74.4) | 37.8 (37.4–38.2) | 62.2 (61.8–62.6) | 41.8 (41.2–42.5) | 58.2 (57.5–58.8) |
| **Race/skin color** | | | | | | |
| White | 53.7 (52.9–54.5) | 53.7 (53.1–54.3) | 46.9 (46.3–47.6) | 47.2 (46.6–47.7) | 43.6 (42.6–44.6) | 45.5 (44.5–46.5) |
| Black/brown | 46.3 (45.5–47.1) | 46.3 (45.6–46.9) | 53.1 (52.4–53.7) | 52.8 (52.2–53.4) | 56.4 (55.3–57.4) | 54.5 (53.5–55.5) |
| **Marital status** | | | | | | |
| Single | 83.3* (82.7–83.9) | 11.0* (10.7–11.3) | 58.1* (57.4–58.8) | 19.8* (19.4–20.1) | 61.5* (60.6–62.4) | 18.7* (18.1–19.4) |
| Married | 16.7* (16.1–17.3) | 89.0* (88.7–89.2) | 41.9* (41.2–42.6) | 80.2* (79.8–80.6) | 38.5* (37.5–39.4) | 81.3* (80.6–81.9) |
| **Educational level** | | | | | | |
| Illiterate/ Never studied | 20.1* (19.5–20.7) | 15.9* (15.4–16.3) | 13.2* (12.8–13.6) | 12.9* (12.6–13.3) | 7.8* (7.4–8.3) | 6.7* (6.4–7.1) |
| 1–7 | 37.5* (36.8–38.2) | 40.2* (39.7–40.8) | 32.6* (32.1–33.2) | 35.0* (34.4–35.5) | 32.2* (31.3–33.0) | 33.5* (32.6–34.3) |
| 8–12 | 30.2* (29.6–30.9) | 33.4* (32.8–33.9) | 39.4* (38.8–40.0) | 38.9* (38.4–39.4) | 38.6* (37.7–39.5) | 41.6* (40.6–42.5) |
| > 12 | 12.1* (11.5–12.6) | 10.5* (10.1–10.9) | 14.7* (14.2–15.3) | 13.2* (12.7–13.6) | 21.4* (20.4–22.4) | 18.2* (17.2–19.3) |
| **Household** | | | | | | |
| **Microregions** | | | | | | |
| North | 6.1* (5.8–6.5) | 7.1* (6.6–7.7) | 7.1* (6.9–7.4) | 7.4* (7.2–7.6) | 7.3* (6.9–7.7) | 7.2* (6.9–7.6) |
| Northeast | 26.4* (25.8–27.1) | 25.6* (25.2–26.0) | 26.9* (26.4–27.5) | 25.8* (25.4–26.2) | 28.1* (27.3–29.0) | 24.3* (23.6–25.0) |
| Midwest | 7.3* (6.9–7.6) | 7.3* (7.1–7.5) | 7.6* (7.4–7.9) | 7.7* (7.5–7.9) | 7.2* (6.7–7.6) | 8.2* (7.7–8.7) |
| Southeast | 45.3* (44.6–46.1) | 44.0* (43.5–44.5) | 43.4* (42.7–44.0) | 43.6* (43.2–44.1) | 41.6* (40.6–42.6) | 45.1* (44.1–46.0) |
| South | 14.8* (14.2–15.4) | 16.0* (15.7–16.3) | 14.9* (14.5–15.4) | 15.5* (15.2–15.8) | 15.7* (15.1–16.4) | 15.2* (14.6–15.8) |
| **Household area** | | | | | | |
| Urban | 91.9* (91.2–92.5) | 81.6* (80.6–82.6) | 91.8* (91.3–92.3) | 82.1* (81.2–82.9) | 89.9* (89.2–90.5) | 83.6* (82.9–84.2) |
| Rural | 8.1* (7.5–8.8) | 18.4* (17.4–19.4) | 8.2* (7.7–8.7) | 17.9* (17.1–18.8) | 10.1* (9.5–10.8) | 16.4* (15.8–17.1) |
| **People in the household** | | | | | | |
| 1–2 | 44.5* (43.7–45.2) | 23.4* (23.0–23.7) | 43.5* (42.9–44.2) | 34.5* (34.1–35.0) | 46.6* (45.6–47.5) | 36.9* (36.1–37.8) |
| 3–5 | 46.7 * (46.0–47.4) | 64.6* (64.2–65.0) | 50.0* (49.4–50.6) | 59.2* (58.7–59.6) | 47.2* (46.2–48.1) | 58.1* (57.3–58.9) |
| > 5 | 8.8 * (8.5–9.2) | 12.0* (11.7–12.3) | 6.4* (6.1–6.7) | 6.3* (6.1–6.5) | 6.2* (5.8–6.7) | 5.0* (4.7–5.3) |
| **Presence of children under 5 years old** | | | | | | |
| Yes | 19.9* (19.4–20.5) | 29.9* (29.5–30.4) | 19.0* (18.5–19.4) | 21.7* (21.3–22.1) | 15.2* (14.6–15.9) | 17.5* (16.9–18.1) |

*(Continued)*

**Table 1.** (Continued)

| | PNAD[1] 2004 | | PNAD[1] 2013 | | POF[2] 2018 | |
|---|---|---|---|---|---|---|
| | Women % (CI 95%) | Men % (CI 95%) | Women % (CI 95%) | Men % (CI 95%) | Women % (CI 95%) | Men % (CI 95%) |
| No | 80.1* (79.5–80.6) | 70.1* (69.6–70.5) | 81.0* (80.6–81.5) | 78.3* (77.9–78.7) | 84.8* (84.1–85.4) | 82.5* (81.9–83.1) |
| **Monthly per capita household income (quintile)** | | | | | | |
| 1st | 4.8* (4.5–5.1) | 5.0* (4.7–5.3) | 19.6* (19.1–20.1) | 18.9* (18.5–19.3) | 17.9* (17.2–18.6) | 15.2* (14.6–15.8) |
| 2nd | 19.5* (18.9–20.0) | 20.7* (20.3–21.2) | 19.6* (19.2–20.1) | 19.0* (18.6–19.4) | 18.8* (18.1–19.6) | 17.2* (16.6–17.8) |
| 3rd | 28.0* (27.3–28.6) | 26.5* (26.0–26.9) | 21.2* (20.7–21.7) | 20.1* (19.8–20.5) | 18.9* (18.2–19.7) | 20.3* (19.6–20.9) |
| 4th | 20.8* (20.3–21.4) | 22.2* (21.8–22.6) | 19.7* (19.2–20.2) | 21.1* (20.7–21.5) | 20.6* (19.8–21.4) | 21.4* (20.7–22.1) |
| 5th | 26.9* (26.1–27.7) | 25.6* (25.0–26.2) | 19.9* (19.3–20.5) | 20.9* (20.3–21.5) | 23.8* (22.7–24.8) | 25.9* (24.8–27.0) |

* Statistical significance (p<0.05). Chi-squared test. [1]PNAD (*Pesquisas Nacionais por Amostras de Domicílios*): Brazilian National Households Sample Surveys; [2]POF (*Pesquisa de Orçamentos Familiares*): Household Budget Survey.

of these factors. Hence, these findings are innovative, as they strongly suggest that an intersectional lens is needed to better understand how to prevent or mitigate FI through a social determinants of health approach.

The Brazilian concept of FNS was approved at the Brazilian 2nd National FNS Conference in 2004 and considers the interactions between dimensions of food security, food safety and integration of the human process, including the socioeconomic and cultural aspects expressed by local culture and food heritage [43,44]. Since then, investments in FNS policies have increased, leading to a strong reduction in social inequities in Brazil [45,46]. This study shows that whereas the FI gap by gender narrowed between 2004 and 2013, it subsequently increased by 2018, likely because of a significant reduction in social investments since 2016. This finding indicates that the FI gender gap is a sensitive indicator of the social inequities faced by women.

In recent years, there has been a significant increase in the percentage of female-headed households in several countries around the world, in which, Latin America and the Caribbean represent the region with the highest proportion of households headed by women (36.2%) [21,47]. In Brazil, this trend has also been reported with the percentage of female household headship increasing gradually over time reaching almost half of Brazilian households nowadays [48,49]. Changes in cultural patterns, such as the decline of marriage, the increase in consensual unions, and a greater number of divorce or separation, in addition to the greater female participation in the labor market and women's higher education are factors likely explaining, at least in part, the trend in the increase in FHHs prevalence [21,47–51]. Changes to the question to identify the head of household could also explain the growth of female headship reported, however the questions about the head of the household were asked the same way to the respondents across surveys and across time.

Consistent with our findings, previous studies conducted in African, Asian, and Latin American countries have shown a higher risk of being poor and reporting FI in FHHs than in MHHs [12,15,52–55]. These findings are consistent with the hypothesis that women disproportionately experience hardships as a result of economic deprivation and other social vulnerabilities, supporting the concept of 'feminization of poverty' [13,56].

## PNAD 2004

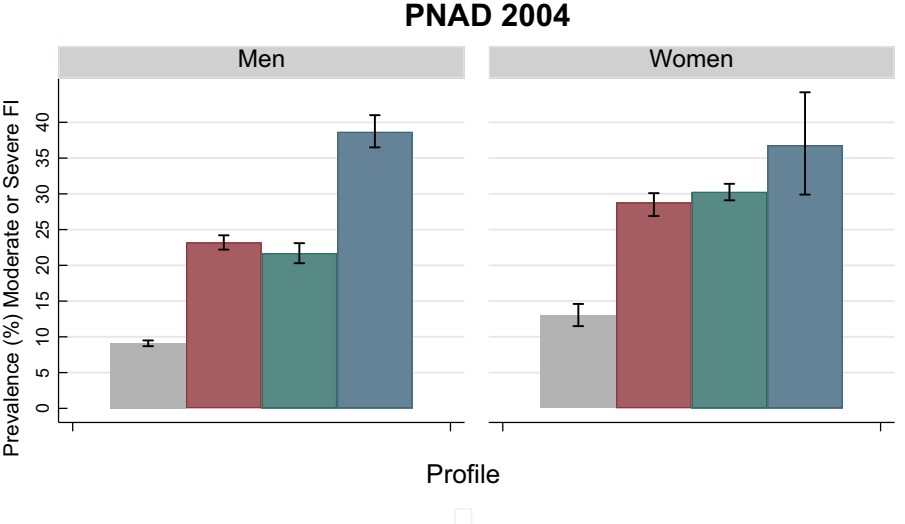

## PNAD 2013

## POF 2018

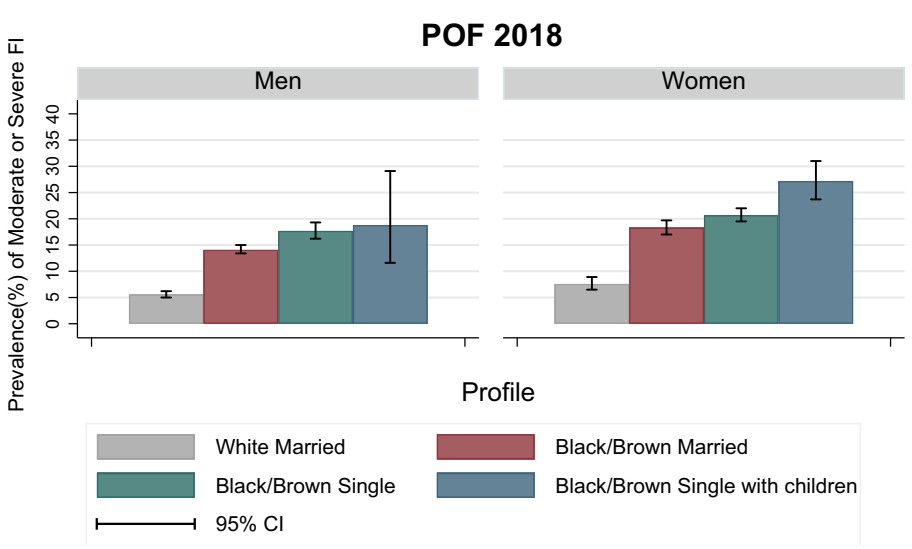

**Fig 2. Prevalence (%) of moderate/severe FI in Brazilian households for subgroups according to the sex of reference person and survey year.** Brazil, 2004, 2013 and 2018.

Previous studies have documented a strong gender-related income gap. The gender roles expected by societies across the globe have assigned women the responsibility for family care-giving, including the feeding of the household, food growing, acquisition and preparation. These expectations require women to spend substantial amounts of time every day on a series of tasks, including meal planning, monitoring the supply of household provisions, food shopping, cooking, and cleaning, while also providing their children with the emotional needs for love, support, and security [7,8]. Because of these domestic activities, women also work more hours per day than men, especially in low- and middle-income countries [13,14,52].

Due to their vulnerable position in society, women are forced to choose poorly paid jobs, often informal work, in the context of little time available for holding formal jobs, lower formal education levels, and lack of time for leisure, which directly impacts their quality of life [11,14]. Additionally, they usually receive lower salaries than men for similar jobs, and they are placed in lower positions compared to men with equal skills, in addition to the lack of recognition or payments for their care roles inside the household [7,57]. Our findings show that

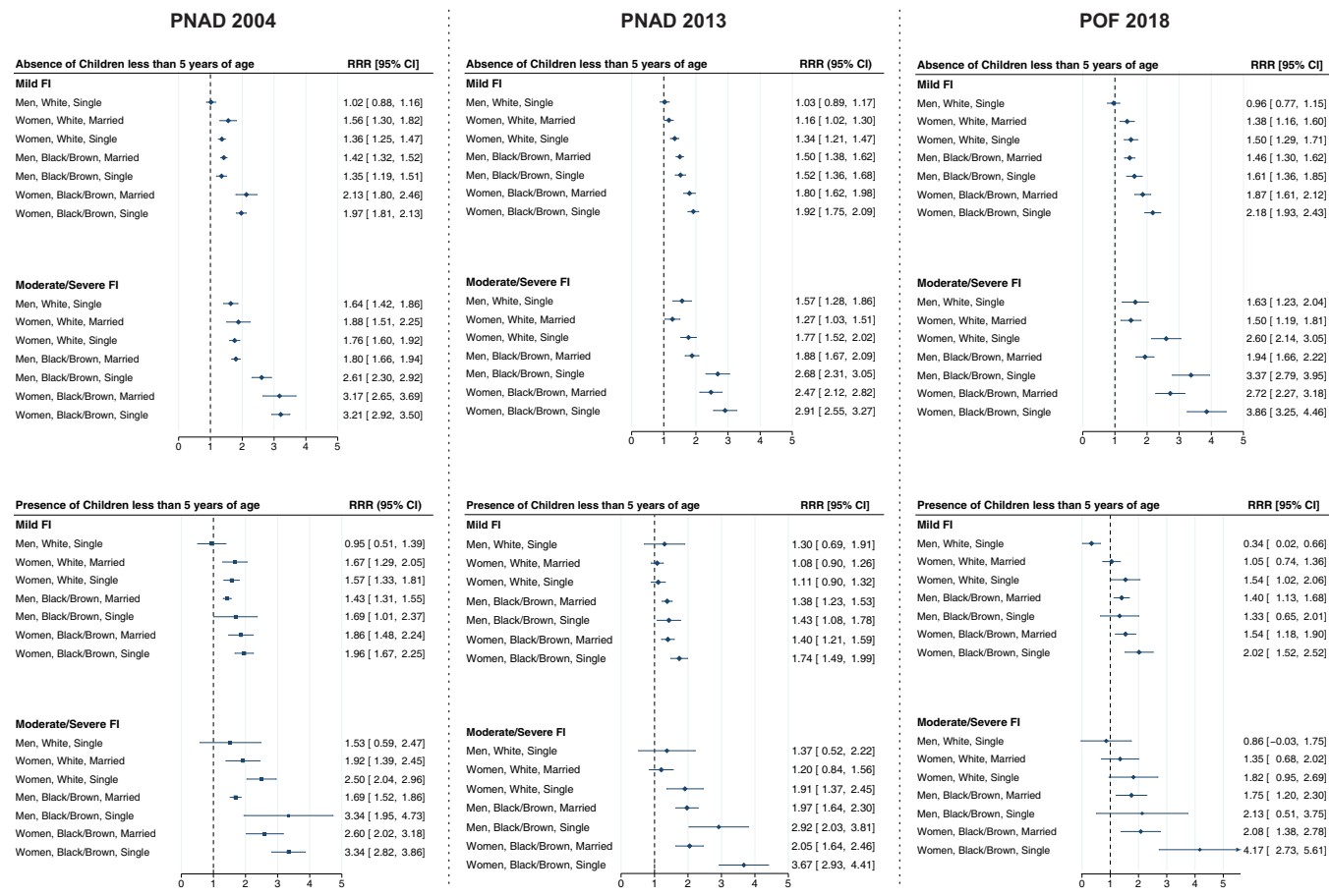

**Fig 3. Relative Risk ratio (RRR) of profiles of reference person of households with sex, race and marital status by food insecurity (FI) levels stratified by presence of children under 5-years old on the household Brazil, 2004, 2013 and 2018.** *Note*: Adjusted for region, area of the household, resident's number and income.

despite progress in women's education, gender-based income persists in Brazil, which is not surprising, as female-headed households are much more likely to live in poverty.

This whole scenario has an impact on the lower income of households headed by women compared to those headed by men. It should also be noted that although income is a key driver of access to the social determinants of FI [12,16], it has some limitations as an indicator of poverty, especially in FHHs, since women face many social constraints (mentioned above) and more difficulties lifting themselves out of poverty [13,52].

Using income as the only indicator of poverty also ignores the nonmonetary resources through which people fulfill their survival needs, such as the "social capital" generated among networks of kin, friends, and neighbors [10,56,58]. In countries with high levels of social inequities, such as Brazil, this gender bias is further exacerbated by the lack of adequate social protection and governmental support for women.

However, some researchers have been questioning the concept of 'feminization of poverty' in the context of FHH [56,59,60]. Medeiros and Costa [61] have argued that this concept does not necessarily imply an absolute worsening in poverty among women within the household but rather that within the household hierarchy, they are more affected than men by poverty-related conditions such as FI. For this reason, poverty in FHHs is not necessarily a proxy for poverty among women per se but rather an explanation about how poverty affects different household members [61]. In our study, the unit of analysis was the 'female-headed household' and not 'women' per se. Hence, the household head indicators were expected to represent the collective FI experiences of all household members, including men, women, and children [61].

Chant [52,56,58] also suggested that women's capacity to command and allocate resources can be more important than the current resource base in their households and that the relationship between women's access to material resources and female empowerment is not straightforward. The author underscores the higher vulnerability of women who live in MHHs because in these households, men control the resources, and they can make them less available to other household members, a situation that could lead to "secondary poverty" among women and children.

In our study, we also found race/skin color-related inequities for moderate/severe FI that further compounded gender-related inequities. Interestingly, black/brown men had higher chances of reporting moderate/severe FI than white women, which indicates how race/skin color plays a significant role in marginalizing people and increasing their vulnerability to FI regardless of gender. This finding suggests that structural racism reflects itself in an increased risk of FI in Brazil [41,62]. Indeed, findings clearly show that black women-headed households were the most vulnerable to moderate/severe FI compared to all other types of households based on the combination of different socioeconomic and demographic characteristics [62,63]. This is likely the result of the intersection between sexism and racism that subjects black women to additional social barriers and structural discrimination that white women do not face [25,26]. Hence, black women face obstacles to their autonomy related to social oppression. These dimensions range from the lack of power of choice and access to healthy foods, to the lack of access to information on healthy eating and knowledge of local products, to the absence of a food environment in their communities favorable to adequate choices [63].

Having at least one child under 5 years old in the household and not being married or cohabiting with a partner at the time of the interview were profiles related to a substantially increased risk of moderate/severe FI for some profiles. We also found that households headed by single white men were less likely to experience FI than those headed by married white men. In contrast, among black/brown men- and women-headed households, being single was a risk factor perhaps because living in households where the reference person was married may have allowed for dual earnings from the couple and the resulting higher income [13].

Consistent with our findings, a study conducted in Nepal also found that married women were 20% less likely to experience household FI than their unmarried counterparts, even after accounting for race/ethnicity and women's education level [64]. However, it is important to acknowledge that living with a partner in a food insecure household may increase the risk of intimate partner violence against women, although it is also possible that women are single as a result of domestic violence [65,66].

Regarding the presence of young children in the household, our findings show that households headed by black/brown single women with at least one child younger than 5 years old may experience more severe levels of FI. The increase in risk of FI in households with children is probably because children may require more household expenses related to their food, health and education needs without usually bringing income to the household [67]. Single mothers may also experience large fluctuations in their household income and become even more vulnerable to poverty and FI [13].

The presence of children in the household can also contribute to a higher likelihood that women will engage in very low paying informal work and will not be able to work in the formal labor force so they can fulfill their care responsibilities at home, especially because of the lack of affordable and safe day care centers for their children [7,8,13,20,68].

However, it should be emphasized that black/brown married women had the opposite experience than single women. An additional explanation for this finding, in addition to the dual earnings of the couple, may be that Brazil's government provides social assistance to families in poverty through conditional cash transfer programs, with benefits increasing as the number of children living in the household grows, which highlights the importance of these programs for these families [69].

It is important to note that the COVID-19 pandemic has worsened FI worldwide, including the widening of the gender gap, especially in lower income countries such as Brazil [1,70]. The pandemic has also contributed to increasing unemployment and reducing family income, aggravating the political, social, and economic crisis of recent years in Brazil [46].

As consequence, the last surveys in the country showed an increase in the FI households in the country in which almost 33.1 million Brazilians faced hunger in the early 2022, 14 million more than in 2018 [45]. Recently, the Brazilian Research Network on Food Sovereignty and Security (*Rede PENSSAN*) released data on gender and race intersectionality in the FI landscape during the COVID-19 pandemic. The results have reinforced that, as in the pre-pandemic period, households headed by black women, regardless of their level of education, have shown lower rates of food secure households and higher vulnerability to FI [71]. Our analytical methodology should be applied to the Rede PENSSAN data to understand how the intersectionality factors have affected FI risks during the pandemic era.

In addition, since 2014, the Brazilian food and nutrition security budget has been reduced since the government eliminated FNS policies and programs that were mitigating FI risks in the most vulnerable households, affecting FNS policies and programs in the country [45,46]. Further, The Brazilian National System of Food and Nutrition Security, created in 2006, has been facing large budget cuts since 2016, similar to many programs aimed at improving FNS, such as conditional cash transfer programs [45]. As a result of the recent election of a new president with strong motivation to address social inequities and food insecurity, this situation may change, so it is crucial to continue monitoring FI time trends in Brazil following the intersectional lens that we propose.

Although our repeated cross-sectional analysis study design is robust, it has some limitations that need to be acknowledged. First, the PNAD and POF surveys have cross-sectional designs, which restricts our ability to draw causal inferences. Second, we were not able to analyze the proportion of income coming from social protection programs such as the *Bolsa*

*Família* cash transfer program or other programs providing benefits to low-income households. Additionally, because of sample size limitations, the study was unable to include households in which the respondent self-identified as "yellow" or belonging to an indigenous group.

Despite these limitations, our study had several strengths, including the tracking of FI in Brazil over a 14-year period with highly comparable nationally representative surveys using an intersectional approach during a period of time when there were drastic changes in social protection policies and investments in the country.

## 5. Conclusions

Our findings indicate that characteristics of the heads of the household, such as gender, race/skin color, marital status, and the presence of young children in the household, are interconnected factors that contribute to an increased risk for household FI in Brazil. Thus, there is a clear need for intersectional policies that address these systemic issues to improve the FNS in the country.

Such policies should be grounded in a deep understanding of the intersectional nature of these issues, recognizing that gender and racial disparities intersect with economic inequality, geographic location, and other factors that affect access to food. Therefore, policymakers should be presented with data disaggregation by sex, race/skin color, household composition and other social determinants to improve the codesign, implementation, evaluation and scaling up of effective policies that take into account the most vulnerable members of society. We also encourage further studies focused on assessing additional food insecurity issues, such as measures that identify if and how different members of the same household experience FI to understand how to ensure that FS policies address the needs of the most vulnerable within the households, including, women, black/brown people, and young children.

Effective social protection policies centered on gender equality, antiracism, and increased attention to families with infants, young children and youth can stimulate economic growth, reduce poverty, and substantially improve household food security in Brazil. The reestablishment of the National Food and Nutrition Security Council by the newly elected government in early 2023, along with the findings of this study, call for considering the reestablishment of relevant ministries, including the Ministry of Women and the Ministry of the Racial Equity, in addition to restoring funding to the Ministry of Social Development, Assistance, Family and Fight against Hunger. These efforts are needed steps toward the mitigation of FI through a new intersectional perspective in Brazil.

## Supporting information

**S1 Table. Prevalence (%) of moderate/severe food insecurity by gender of reference person of the household at the national level and in the different macroregions of Brazil (2004, 2013 and 2018).** FI: Food insecurity. [1]PNAD (*Pesquisas Nacionais por Amostras de Domicílios*): Brazilian National Households Sample Surveys; [2]POF (*Pesquisa de Orçamentos Familiares*): Household Budget Survey.
(DOCX)

**S2 Table. Logistic Models for relationship between moderate/severe Food Insecurity and profiles of reference person of households with sex, race and marital status stratified by region. Brazil, 2004, 2013 and 2018.** *Statistical significance ($p<0.05$). Multinomial Logistic regression. Model adjusted for area of the household, presence of children under 5 years old, resident's number and income. aRRR: adjusted relative risk ratio. [1]PNAD (*Pesquisas Nacionais por Amostras de Domicílios*): Brazilian National Households Sample Surveys; [2]POF (*Pesquisa*

*de Orçamentos Familiares*): Household Budget Survey.
(DOCX)

**S3 Table. Prevalence (%) of Food Security/Food Insecurity status of each profile of sex, race/skin color and marital status of the reference person of the household.** Brazil, 2004, 2013 and 2018. FS: Foos Security; FI: Food Insecurity. [1]PNAD (*Pesquisas Nacionais por Amostras de Domicílios*): Brazilian National Households Sample Surveys; [2]POF (*Pesquisa de Orçamentos Familiares*): Household Budget Survey.
(DOCX)

**S4 Table. Multinomial Logistic Models crude estimates for relationship between Food Security and Food Insecurity levels and profiles of reference person of households with sex, race and marital status stratified by presence of children under 5-years old on the household.** Brazil, 2004, 2013 and 2018. *Statistical significance (p<0.05). Multinomial Logistic regression. Models adjusted for region, area of the household, resident's number and income. cRRR: crude estimates of relative risk ratio. [1]PNAD (*Pesquisas Nacionais por Amostras de Domicílios*): Brazilian National Households Sample Surveys; [2]POF (*Pesquisa de Orçamentos Familiares*): Household Budget Survey.
(DOCX)

## Acknowledgments

The authors would like to thank Luis Paulo Vidaletti Ruas and Cíntia Borges, from the International Center of Equity in Health, for their invaluable assistance with the figures for this paper.

## Author Contributions

**Conceptualization:** Lissandra Amorim Santos, Rafael Pérez-Escamilla, Rosana Salles-Costa.

**Data curation:** Lissandra Amorim Santos, Camilla Christine de Souza Cherol.

**Formal analysis:** Lissandra Amorim Santos, Camilla Christine de Souza Cherol.

**Funding acquisition:** Rosana Salles-Costa.

**Project administration:** Rosana Salles-Costa.

**Resources:** Rosana Salles-Costa.

**Supervision:** Rafael Pérez-Escamilla, Rosana Salles-Costa.

**Writing – original draft:** Lissandra Amorim Santos, Aline Alves Ferreira.

**Writing – review & editing:** Lissandra Amorim Santos, Rafael Pérez-Escamilla, Camilla Christine de Souza Cherol, Aline Alves Ferreira, Rosana Salles-Costa.

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
