## [Decision Letter · Decision Letter 0]

16 May 2023

PGPH-D-23-00533

Household food insecurity in Brazil: the intersection of gender, race, and household composition

Dear Dr. Perez-Escamilla,

Thank you for submitting your manuscript to PLOS Global Public Health. After careful consideration, we feel that it has merit but does not fully meet PLOS Global Public Health’s publication criteria as it currently stands. Therefore, we invite you to submit a revised version of the manuscript that addresses the points raised during the review process.

We look forward to receiving your revised manuscript.

Kind regards,

Godfred Boateng

Academic Editor

Journal Requirements:

Additional Editor Comments (if provided):

Reviewers' comments:

Reviewer's Responses to Questions

**Comments to the Author**

1. Does this manuscript meet PLOS Global Public Health’s publication criteria? Is the manuscript technically sound, and do the data support the conclusions? The manuscript must describe methodologically and ethically rigorous research with conclusions that are appropriately drawn based on the data presented.

Reviewer #1: Yes

Reviewer #2: Yes

2. Has the statistical analysis been performed appropriately and rigorously?

Reviewer #1: Yes

Reviewer #2: Yes

3. Have the authors made all data underlying the findings in their manuscript fully available (please refer to the Data Availability Statement at the start of the manuscript PDF file)?

Reviewer #1: Yes

Reviewer #2: Yes

4. Is the manuscript presented in an intelligible fashion and written in standard English?

Reviewer #1: Yes

Reviewer #2: Yes

5. Review Comments to the Author

Reviewer #1: The authors aim to document trends in household food insecurity in Brazil across time and identify inequities by demographic characteristics. They find, using appropriate and robust analytic methodologies, that female-headed households (particularly those whose household heads have black/brown skin and are single) are at greatest risk of experiencing food insecurity relative to other groups. These results are timely and have the potential to inform the development and implementation of policies and programs that advance national nutrition goals. Below are several suggested revisions to strengthen the manuscript.

1. Abstract: Consider opening with 1-2 sentences about the why this study was conducted and the knowledge gaps it aims to address.

2. Abstract: Remove causal language. For instance, replace “increased the risk of” with “was associated with higher risk of”.

3. Abstract: If possible, add additional methodological details, particularly information about how households were classified into one of the eight gender-race-marital classes.

4. Introduction: To attract a wider readership who may not be familiar with the concept of food insecurity, please add greater detail about this construct. Specifically, it would be helpful to include a definition of food insecurity as well as findings demonstrating its negative impacts on well-being. Such information would help showcase why food insecurity is an important indicator to track and intervene upon for improving public health.

5. Introduction: The authors cite relevant literature to support the notion that female-headed households have less access to financial capital and thus may be at greater risk of experiencing food insecurity. I think it would also be compelling to note that females may also have lower social capital, experience substantial barriers to accessing and benefiting from affordable foods (e.g., engaging in transactional sex to acquire preferred foods [see Fiorella, K. J. et al. A review of transactional sex for natural resources: Under-researched, overstated, or unique to fishing economies?]), or alter their behaviors to buffer other household members from the negative consequences of food insecurity [see Maxfield, A. Testing the theoretical similarities between food and water insecurity: Buffering hypothesis and effects on mental wellbeing.].

6. Introduction: Most of the introduction describes the factors that plausibly increase the risk of food insecurity among female-headed, related to male-headed, households. It would be useful to provide similar explanations for the other characteristics under investigation, namely race/skin color and marital status. The authors should also incorporate relevant theoretical evidence about the importance and development of intersectional theory. I encourage the authors to review a recent paper that provides a concise history of the concept of intersectionality and its utility for public health [Shah, S. H. et al. Variations in household water affordability and water insecurity: An intersectional perspective from 18 low- and middle-income countries.].

7. Introduction: The authors clearly state why patterns of food insecurity should be examined in Latin American and Caribbean countries but do not justify why Brazil should be studied in particular. It may be helpful to include empirical evidence that food insecurity is a prevalent issue in Brazil and demonstrated to undermine health and well-being.

8. Methods: Were there any additional inclusion/exclusion criteria? For instance, what was the minimum age to participate?

9. Methods: Were all surveys conducted during the same time of year/season? Could variations in the timing of surveys account for some of the differences observed between survey waves?

10. Methods: What language(s) were surveys conducted in? Were any of the survey modules piloted before broad implementation to ensure that items were understood as intended?

11. Methods: Describe how food insecurity scores were scaled to ensure comparability between households with and without children.

12. Methods: It would be helpful to provide the theoretical range of food insecurity scores as well as the cut-points for determining different levels of food insecurity.

13. Methods: Consider reporting the internal validity of the food insecurity scale across sampling waves.

14. Methods: What proportion of respondents had missing data? How were missing data handled? Either describe how missing data are unlikely to meaningfully influence results or conduct sensitivity analyses to estimate the potential bias introduced from complete-case analysis.

15. Methods: Is there empirical evidence to support that the findings from the two different surveys are comparable? Additional justification is needed given that this is a critical assumption of the study.

16. Methods: What proportion of individuals identified as having black compared to brown skin? What is the justification for combining these two categories?

17. Methods: How was urban/rural household location determined?

18. Methods: When determining which confounders to include in the model, why was a data-driven, as opposed to a theoretically driven, approach used? As noted in numerous epidemiological reports, such approaches can introduce bias [e.g., VanderWeele, T. J. Principles of confounder selection]. Consider running several models as a sensitivity analysis to evaluate whether findings are consistent across model specification.

19. Methods: What information led the authors to conclude that having children less than 5 in the household was an effect measure modifier? Please include these details in the main text of the manuscript.

20. Methods: The authors note that “multinomial logistic regressions were conducted for each of the three surveys to estimate the prevalence of mild and moderate/severe FI across the intersection of gender”, but then report odds ratios. Is it possible to fit generalized linear models to obtain prevalence differences between groups, which are more valuable for informing public health decisions than relative measures [see Poole, C. On the Origin of Risk Relativism]?

21. Methods: Please describe how informed consent and ethical approvals were obtained within the parent studies.

22. Methods: The use of multinomial logistic regression is sensible. I wonder, however, if ordinal logistic regression was considered given that the outcome is ordinal.

23. Methods: Note why married white men were selected as the referent category.

24. Results: In Figure 1, please consider including error bars around the point estimates.

25. Results: In Table 1, “qui-squared” should be “chi-squared”. Also, make clear in the caption that gender refers to that of the household head, not necessarily that of the survey respondent.

26. Results: Consider also presenting aggregate food insecurity data across time (i.e., not divided by subgroups) to understand population-level trends.

27. Results: I am finding it difficult to interpret Figure 2. Please consider providing more information in the legend to make clear what each bar represents. For instance, one bar is meant to represent “gender and race”, but which gender and race category?

28. Results: When comparing changes across time, it might be useful to include estimated percentage-point changes to support statements like “men had the greatest increases across time”.

29. Results: At line 246, would it be more accurate to replace “likelihood of FI” with “prevalence of FI”?

30. Results: Table 2 includes a lot of great information, but it is difficult to interpret. Consider creating a coefficient plot that includes point estimates for each “person type” across the years. This would be useful to track temporal trends and highlight which groups are consistently at greatest risk for experiencing food insecurity.

31. Results: Please provide information about the percent of households categorized into each of the eight categories across time (e.g., what percent of households were headed by white single men at sampling period?). Currently, such information cannot be derived from Table 1.

32. Discussion: The Discussion is strong and engages deeply with relevant literature. I only have a few suggestions on how to potentially increase its impact. For instance, it may be helpful to describe why there were such dramatic shifts in the percentage of households headed by females across sampling waves. Is there evidence of demographic shifts or changes in gender norms across the 14-year study period? Did the surveys include probes for soliciting information about the gender of the household head that potentially changed across survey waves?

33. Discussion: At line 304, consider listing the specific policy and programmatic changes that were made. For instance, was support for and benefits included in all social programs reduced, or specifically those related to nutrition?

34. Discussion: Provide more details about the studies described and cited at line 307. Which regions of the world were these studies conducted in?

35. Discussion: Consider describing the utility of measuring individual food insecurity in future studies to understand intra-household variations in food insecurity experiences.

Reviewer #2: The paper aims at analysing the intersection of gender, race and the presence of children in the household with the household food insecurity, in Brazil in various years from 2004 and 2018.

The issue of food insecurity is of great interest for global public health.

The authors treat the literature in the Introduction and Discussion Sessions.

The manuscript is written very clearly and it is accessible also to non-specialists.

However, I have some comments.

A first limitation of the study resides in the data that dates back to 2018. Since the global outlook of food insecurity has been affected by notable changes in recent years, it would be interesting to refer the analysis to more recent data, or at least to explain to the readers why this has not been done.

However, the study has been able to analyse food insecurity in Brazil over a 14-year period with highly comparable nationally representative surveys.

The manuscript can be better organized in the following way:

- Section 2 should be named “Data” rather than “methods”, since it deals with the data used in the study

- Section 2.5 could be numbered “3” as it refer to the empirical methodology

- Section 2.6 can be put as a footnote in section 2.1

I should invite the authors to put particular attention to the labels about “race”. I suggest to avoid overlapping the concept of race with that of skin colour without explanation, since this is not universally accepted. I think the authors could write only “race” specifying (starting from the abstract) that they identify “race” with the self-stated colour of the skin (as specified in Section 2.3). More in detail, I suggest to specify the first time the authors use the term “race” that they actually mean the “self-identified phenotype” (both in the abstract and in the introduction). I also think that the label “mixed-race” is misleading, since it refers to brown/black phenotypes. I suggest to find a more accurate label.

6. PLOS authors have the option to publish the peer review history of their article (what does this mean?). If published, this will include your full peer review and any attached files.

**Do you want your identity to be public for this peer review?** For information about this choice, including consent withdrawal, please see our Privacy Policy.

Reviewer #1: **Yes: **Joshua D. Miller

Reviewer #2: No

---

## [Decision Letter · Decision Letter 1]

3 Aug 2023

Gender, racial and household composition explain inequities in household food insecurity in Brazil

PGPH-D-23-00533R1

Dear Dr. Perez-Escamilla,

We are pleased to inform you that your manuscript 'Gender, racial and household composition explain inequities in household food insecurity in Brazil' has been provisionally accepted for publication in PLOS Global Public Health.

Best regards,

Godfred Boateng

Academic Editor

I reviewed your paper with great interest and find it to advance our understanding of the intersectionality of food insecurity and other social determinants of health. The analytical approach was well designed and executed, and the findings presented in a logical manner. This paper is novel for the use of data from Brazil to show how other factors such as race/skin color, marital status of the household head, and the presence of children at home, may further worsen the prevalence of moderate/severe food insecurity in female headed households compared to male headed households. Having addressed to my satisfaction the reviewers comments, I recommend this paper for publication.

Reviewer's Responses to Questions

**Comments to the Author**

1. If the authors have adequately addressed your comments raised in a previous round of review and you feel that this manuscript is now acceptable for publication, you may indicate that here to bypass the “Comments to the Author” section, enter your conflict of interest statement in the “Confidential to Editor” section, and submit your "Accept" recommendation.

Reviewer #1: All comments have been addressed

2. Does this manuscript meet PLOS Global Public Health’s publication criteria? Is the manuscript technically sound, and do the data support the conclusions? The manuscript must describe methodologically and ethically rigorous research with conclusions that are appropriately drawn based on the data presented.

Reviewer #1: Yes

3. Has the statistical analysis been performed appropriately and rigorously?

Reviewer #1: Yes

4. Have the authors made all data underlying the findings in their manuscript fully available (please refer to the Data Availability Statement at the start of the manuscript PDF file)?

Reviewer #1: Yes

5. Is the manuscript presented in an intelligible fashion and written in standard English?

Reviewer #1: Yes

6. Review Comments to the Author

Reviewer #1: I commend the authors for their thoughtful responses to the reviewers' comments and rigorous analyses. This manuscript will be a great addition to the current food security literature. Below are a few minor suggestions to consider before publication.

1. The Results section (Lines 315-337, 378-384) contains causal language that should be eliminated. For instance, the authors state that "Figure 2 shows how the stepwise addition of the household head characteristics gender, race/skin color, marital status, and the presence of children in households led to a stepwise increase in the prevalence of moderate/severe FI in households". Instead, the authors could note that "Figure 2 shows how concurrent consideration of gender, race/skin color, marital status of the household head, as well as the presence of children in households, provides a more nuanced understanding of who, exactly, is most likely to be food insecure. For instance, relative to households headed by married white men..."

2. I appreciate that the authors reported percentage point differences to help illustrate how food insecurity varied across groups and within groups across time. Consider adding the 95% confidence intervals for each point estimate so readers can see the range of plausible values.

3. There is a small issue with Figure 1: the y-axis for the bottom graph extends into the middle graph, creating "floating" 40% and 50% values.

4. It is currently difficult to discern the different categories presented in Figure 2. Consider using additional colors or patterns to help distinguish each.

7. PLOS authors have the option to publish the peer review history of their article (what does this mean?). If published, this will include your full peer review and any attached files.

**Do you want your identity to be public for this peer review?** For information about this choice, including consent withdrawal, please see our Privacy Policy.

Reviewer #1: **Yes: **Joshua D. Miller
